# Murine Fibroblasts and Primary Hepatocytes as Tools When Studying the Efficacy of Potential Therapies for Mucopolysaccharidosis Type I

**DOI:** 10.3390/ijms24010534

**Published:** 2022-12-28

**Authors:** Magdalena Węsierska, Wioletta Nowicka, Anna Kloska, Joanna Jakóbkiewicz-Banecka, Marcelina Malinowska

**Affiliations:** Department of Medical Biology and Genetics, Faculty of Biology, University of Gdańsk, Wita Stwosza 59, 80-308 Gdańsk, Poland

**Keywords:** flavonoids, GAG synthesis, SRT, MPS I

## Abstract

Mucopolysaccharidosis type I (MPS I) is a metabolic genetic disease caused by the deficiency of a lysosomal enzyme involved in glycosaminoglycans (GAGs) degradation. MPS I cells have a constant level of GAG synthesis, but disturbed degradation means that GAGs accumulate progressively, impairing cell metabolism. GAG metabolism can be modulated by flavonoids, and these are being studied as therapeutics for MPS. We have optimised the protocol for obtaining fibroblasts and hepatocytes from the MPS I murine model and characterised the cells for their suitability as an in vitro model for testing compounds with therapeutic potential. Methods: Murine primary hepatocytes and fibroblasts were used as a cellular model to study the effect of genistein, biochanin A, and kaempferol on the modulation of the GAG synthesis process. Flavonoids were used individually as well as in two-component mixtures. There were no statistically significant differences in GAG synthesis levels from cell types obtained from either wild-type or MPS I mice. We also showed that MPS I fibroblasts and hepatocytes store GAGs, which makes them useful in vitro models for testing the effectiveness of substrate reduction therapies. Furthermore, tested flavonoids had a different impact on GAG synthesis depending on cell type and whether they were used alone or in a mixture. The tested flavonoids reduce GAG synthesis more effectively in fibroblasts than in hepatocytes, regardless of whether they are used individually or in a mixture. Flavonoids modulate the level of GAG synthesis differently depending on cell types, therefore in vitro experiments performed to assess the effectiveness of potential therapies for metabolic diseases should be carried out using more than one cell model, and only such an approach will allow for full answering scientific questions.

## 1. Introduction

Mucopolysaccharidosis type I (MPS I, OMIM 252800) is a lysosomal storage disorder (LSD) caused by a deficiency of α-L-iduronidase (IDUA), which is a lysosomal hydrolase involved in glycosaminoglycans (GAG) catabolism. The impaired activity of IDUA causes a progressive accumulation of heparan sulphate (HS) and dermatan sulphate (DS), leading to cellular, tissue, and organ dysfunction. Clinically, MPS I exhibits a wide phenotypic spectrum classified into three subtypes: severe Hurler, attenuated Hurler/Scheie, and mild Scheie [1]. Cellular GAG storage leads to the development of many somatic symptoms, and the severity, as well as progression, depends on the genotype. Patients usually develop an enlarged liver and spleen with a ventral hernia, numerous bone deformities with coarse facial features, hearing and vision problems, and respiratory and cardiac impairment. In the most severe type of disease, the central nervous system (CNS) and cognitive functions are impaired. The range of clinical manifestations was recently reviewed [2]. The wide-ranging severity of clinical symptoms means that early diagnosis and treatment are difficult [3]. There are currently two therapeutic options available for MPS I: hematopoietic stem cell transplantation (HSCT) and enzyme replacement therapy (ERT). Since each of these has significant limitations and is not effective in dealing with all clinical symptoms, treatment methods based on mechanisms other than the delivery of the missing enzyme are being sought [4].

One option for the treatment of LSD was investigated for many years, namely substrate reduction therapy (SRT), which is based on preventing GAG storage by decreasing the level of their biosynthesis and accumulation. Perspectives on current and future SRT were recently revised [3]. Flavonoids, the most vital phytochemicals with diverse bioactivities, are compounds that can potentially be used as active agents in SRTs. These compounds comprise the largest group of natural products in respect of quantity and quality. In terms of their chemical structure, the molecules are commonly occurring polyphenolic secondary metabolites of plants, and their broad biological activities are used in preventing and managing many diseases, such as cancer, bacterial and viral infections, inflammation, and genetic diseases [5,6,7,8,9]. Furthermore, the development of therapeutic strategies for MPS is limited due to the presence of the blood-brain barrier (BBB), which is highly impermeable to large therapeutic molecules such as enzymes used in enzyme replacement therapy (ERT), making neurological symptoms difficult to ameliorate. Alternatively, the low molecular weight molecules used in SRT could potentially readily cross the BBB and provide disease treatment benefits by reducing intracellular GAG synthesis to reduce pathological storage and slow disease progression in affected patients. The evidence suggesting the therapeutic potential of flavonoids in SRT for MPS [10,11] is interesting, but still needs to be verified, as the results of studies to date are often contradictory, and positive effects observed in cell culture studies or using animal models are not confirmed in clinical trials [12].

Most studies on the use of flavonoids in SRT for MPS have focused on genistein, a naturally occurring soy isoflavone, which acts by inhibiting the tyrosine kinase activity of the epidermal growth factor receptor (EGFR) [9]. The therapeutic activity of genistein was first described in 2006 when Piotrowska et al. showed that this compound reduces the level of GAG synthesis and storage in MPS I fibroblasts [13]. Genistein and other flavonoids were subsequently tested in various in vitro [10,11] and in vivo [14,15,16,17] models, and clinical trials [18,19,20,21,22]. Unfortunately, the published results are sometimes ambiguous or contradictory. Moreover, it seems that selecting an appropriate research model may be the most important aspect here, as it has been observed that genistein reduces GAG levels in fibroblasts [13], but increases them in chondrocytes [23]. Another study showed significant adverse effects in MPS I mice after the oral administration of genistein, including reduced skeletal growth and scrotal hernia, presumably due to the high biological activity of genistein and its non-specific effect on many different pathways [17]. Such inconsistent results discourage the use of genistein alone in the treatment of MPS I patients.

Other compounds classified as flavonoids, such as kaempferol, daidzein, biochanin A, formononetin, and glycitein were reported to efficiently inhibit GAG synthesis or reduce their storage in MPS cellular models [10,11,24]. In addition, research has been conducted on the form of administration of flavonoids to increase their effectiveness in modulating GAG synthesis in SRT [25]. Other natural sources of flavonoids that would provide new potential therapeutics for the treatment of MPS are also being sought [26]. It seems that flavonoids applied in mixtures also act more efficiently than when used alone [27,28]. Recent studies found that genistein and kaempferol modulate the expression of genes involved in pathways of GAG synthesis and degradation [27], and genistein has been also shown to induce increased lysosomal biogenesis due to the regulation of transcription factor EB (TFEB) [29].

A combination of two active agents may have increased or reduced effects, and the main reason for using such a combination is to increase the efficacy of the treatment while decreasing its toxicity. The combination index (CI) analysis by Chou-Talalay method [30] is used to determine the type and degree of interaction between the components of the mixture. We recently performed such an analysis, characterising the interactions between genistein and kaempferol concerning cellular and gene expression responses, and evaluating their potential as SRT for MPS I [31].

In our present study, we compared the level of GAG synthesis and storage in fibroblasts and hepatocytes and characterised their response to single flavonoid treatment (genistein, kaempferol, and biochanin A). We also investigated the effect of two-flavonoid mixtures (genistein and kaempferol, genistein and biochanin A) on GAG synthesis, followed by an analysis of interactions between tested flavonoids.

## 2. Results

### 2.1. Characteristics of Murine Cells

Fibroblasts and primary hepatocytes obtained from MPS I and wild-type (WT) mice were morphologically assessed during cell culture at several time points. Fibroblasts migrated from the skin explants, and their significant growth on the culture dish was observed near the tissue, as well as in zones more distant from the explant. No differences in cell morphology were observed between WT and MPS I fibroblasts. Both cell types of fibroblasts were in good metabolic condition and did not show any disturbances in cell division at the early stage of the culture, but their morphology depended upon the extent of confluency. Cells could be passaged until the fourth passage was reached when senescent cells appeared in the culture and the number of viable cells decreased (Figure 1).

Obtaining high yields of viable primary mouse hepatocytes is a complicated and multi-stage process, which significantly limits their experimental use. WT and MPS I hepatocytes can be maintained in culture for about a week. However, a proper experiment should be conducted within three days of starting the culture, as cells show the highest viability during this time. Hepatocytes are cells that do not divide in vitro, and therefore after six days, the culture enters its declining phase, which is manifested by a change in the morphology of cells, loss of viability, and detachment from the bottom of the dish (Figure 2).

We characterised these cells in terms of GAG synthesis and storage to evaluate the usefulness of the obtained cell culture models as a tool for testing potential active agents for substrate reduction therapy in MPS I. We did not observe statistically significant differences in the level of GAG synthesis between WT and MPS I cells in either cell model (Figure 3).

Although there was no statistically significant difference in the kinetics of GAG synthesis between WT and MPS I in either type of cells, they differed significantly in the level of stored GAGs. MPS I fibroblasts stored more GAGs during culturing compared to hepatocytes, reaching the maximum storage level in the fourth week, while a decrease in storage was observed in the sixth week. At this time point, the fibroblasts in a culture entered the declining phase with a high rate of cell death. The GAG level in WT cells declined after the third week of culturing, suggesting that this phenomenon is related to the metabolic state of cells in culture (Figure 4A). The GAG level in primary MPS I hepatocytes was statistically significantly and higher than in WT hepatocytes throughout the entire culture (Figure 4B), with the highest levels observed between 72–96 h of culture. Since primary hepatocytes are cells that do not divide in vitro and appear to dedifferentiate when grown as monolayers, this time point is the most optimal for conducting experiments. The level of GAG storage in MPS I hepatocytes was more stable than in fibroblasts during the culture, but it should be remembered that hepatocytes showed a decrease in viability, and lost their hepatic potential after prolonged culture, which was manifested by a change in morphology (Figure 2).

### 2.2. Effect of Genistein, Biochanin A, and Kaempferol on GAG Synthesis Kinetics

All tested flavonoids inhibited GAG synthesis in murine MPS I fibroblasts and hepatocytes in a dose- and cell-dependent manner (Figure 5). Genistein, when used at higher concentrations (30 and 60 µM), decreases GAG synthesis in a statistically significant manner in both cell models compared to untreated control (Figure 5A). There was also a statistically significant difference in the response to high concentrations of genistein depending on the cell type. Genistein reduced the kinetics of GAG synthesis more in fibroblasts than in hepatocytes (Figure 5A). The most effective modulator of GAG synthesis was kaempferol, which decreased its level at all tested concentrations (15, 30, and 60 µM) compared to the untreated control (Figure 5B). However, as in the case of genistein, the response differed significantly depending on the cell type. GAG synthesis is reduced by about 60% in fibroblasts, and only by about 40% in hepatocytes. The flavonoid that showed the weakest effect on the level of GAG synthesis was Biochanin A, and a statistically significant reduction was observed only in fibroblasts at concentrations of 15 and 60 µM compared to the control (Figure 5C).

### 2.3. Effect of Two Compound Mixtures on GAG Synthesis Kinetics

Mixtures of genistein and kaempferol (G + K), and genistein and biochanin A (G + B) significantly reduced GAG synthesis at final concentrations of 30 and 60 µM in a dose-dependent and cell-dependent manner (Figure 6). A mixture of genistein and kaempferol in the ratio of 4:1, 3:2, 2:3, and 1:4 statistically significantly inhibited GAG synthesis in nearly all experimental setups. However, fibroblasts responded more strongly to the tested compounds, especially at the final concentration of 60 µM. The combination of genistein and kaempferol was also more effective in reducing the level of GAG synthesis in both cell models.

### 2.4. Analysis of Interactions between Flavonoids in Two-Compound Mixtures

Normalised isobolograms were constructed to visualise the interactions between flavonoids in two-component mixtures. A point plotted on the isobolograms below the theoretical line of additivity means that the components of the tested mixture act synergistically, while a point plotted above that line means that the components act antagonistically.

The tested flavonoids used in two-component mixtures of genistein and kaempferol or genistein and biochanin A showed the entire spectrum of interactions (Figure 7). For fibroblasts, genistein, and biochanin A acted synergistically in a 30 µM-mixture, while genistein and kaempferol acted synergistically in a 60 µM-mixture as GAG synthesis inhibitors (Figure 7A). Interactions were different for hepatocytes in the same mixtures. We observed that mixtures of genistein with either biochanin A or kaempferol act antagonistically against the inhibition of GAG synthesis in most of the ratios and final concentrations tested (Figure 7B; with most of the data points plotted outside the isobolograms, denoting remarkably strong antagonism).

The combination index (CI) value was used to quantitatively measure the degree of synergy or antagonism between flavonoids in a mixture. According to CI values, the interactions were defined as synergism (CI: 0.30–0.69), moderate synergism (CI: 0.70–0.89), nearly additive (CI: 0.90–1.09), slight antagonism (CI: 1.10–1.19), moderate antagonism (CI: 1.20–1.44), and antagonism (CI: 1.45–3.3) [30]. CI values for fibroblasts were either higher or lower than 1.0 for the mixtures (CI range, 0.79–1.60 for genistein and biochanin A, and 0.74–2.39 for genistein and kaempferol), indicating either antagonistic (CI > 1) or moderately synergistic (CI < 1) effects on GAG synthesis between flavonoids, depending on the final concentration and ratios used (Figure 8A and Table 1). CI values for hepatocytes were higher than 1.0 for most of the mixtures (CI range, 0.98–46313 for genistein and biochanin A, and 0.74–4.43 for genistein and kaempferol), indicating antagonistic effects on GAG synthesis between flavonoids (Figure 8B and Table 1).

The reduced dose of genistein, biochanin A, and kaempferol within the combinations is favourable for most of the ratios tested against fibroblasts (Figure 9A and Table 1). However, we observed a favourable dose reduction of biochanin A for hepatocytes, when mixed with genistein (Figure 9B, top graph; Table 1) and genistein when mixed with kaempferol (Figure 9B, bottom graph; Table 1). For hepatocytes, the only mixture that showed moderate synergism in inhibiting GAG synthesis was the mixture of genistein and kaempferol at a ratio of 4:1. Using this composition, it is possible to reduce the dose of genistein and kaempferol by about 2.5 and 2.9 times, respectively (Table 1). Interestingly, the combination of genistein and kaempferol was characterised by moderate synergism against fibroblasts but exhibited antagonism against hepatocytes (Figure 10). The mixture of genistein and kaempferol in a total concentration of 30 µM in both cell culture models resulted in antagonistic interactions, but the effect is more pronounced in hepatocytes. Interestingly, for this mixture at the higher concentration of 60 µM, the interactions become synergistic for fibroblasts while remaining antagonistic for hepatocytes (see G + K panel in Figure 10). Interestingly, the interaction spectrum for both cell types is quite the opposite for the mixture of genistein with biochanin A. Compounds in this mixture at a final concentration of 30 µM show synergistic and additive interactions in fibroblasts, and antagonistic interactions in hepatocytes. For fibroblasts exposed to that mixture at a final concentration of 60 µM, the components show antagonistic effects similar to that in hepatocytes (see G + B panel in Figure 10).

## 3. Discussion

Mucopolysaccharidosis type I is a metabolic disease in which the functioning of many organs is impaired. In humans, hepatosplenomegaly is one of the first symptoms to appear during the first six months of life [1]. Similarly, a pathologically-enlarged liver and spleen appear in 8–10 week-old MPS I mice [32]. Most in vitro studies on the effectiveness of potential therapies for MPS I to date. However, have been conducted on fibroblasts and not on hepatocytes or splenocytes [25,33,34,35]. On the one hand, fibroblasts are a good MPS I model because these cells are responsible for maintaining the extracellular matrix composed of GAGs and proteoglycans, they participate in tissue repair and remodelling [36], and are easy to obtain from both patients and animals. On the other hand, the liver is the primary organ in which GAG accumulation occurs, resulting in hepatomegaly, and thus hepatocytes should be the most appropriate cell culture model selected for studying MPS I pathology or potential therapies. The key question is whether these two MPS I cell types perform similarly when the intracellular homeostasis is disrupted due to genetically determined α-iduronidase deficiency. We characterised and compared murine MPS I fibroblasts and primary hepatocytes in terms of GAG synthesis and accumulation during in vitro culture.

Recent data indicates that the embryonic origin, and structural and functional features of the cell, may alter signalling pathways specific to the α-iduronidase-deficient cell [37]. However, here we show, that the overall level of GAG synthesis is comparable between fibroblasts and hepatocytes, as well as between MPS I and WT cells, which indicates that both cell culture models are suitable for in vitro studies of this disease, and the effectiveness of novel molecules for substrate reduction therapy. Moreover, in both cell culture models, the level of GAG accumulation is significantly higher in MPS I compared to WT cells, allowing a good distinction between the two phenotypes. The maximum duration of in vitro culture is undoubtedly shorter for hepatocytes than for fibroblasts (days vs. weeks). However, the GAG content in MPS I hepatocytes remains constant during culture, while in fibroblasts it is noted to be the highest at Week 4 and then decreases around Week 6 of culture. These measurements correspond to the greatest viability and proliferation capacity of both cell types examined by microscopic inspection. Our results, therefore, suggest that the time point for starting experiments investigating the GAG metabolism and storage is crucial and should be carefully planned: experiments with mouse fibroblasts should not be carried out on cultures beyond the fourth passage because at that time cells start to show symptoms of senescence, and experiments with primary hepatocytes should be performed within three days of the time of isolation and the initiation of culture when the cells show the highest viability. Such a scheme should ensure that experiments are carried out when cells are in good physiological condition, and adequately reflect the disease phenotype, so that the results obtained are reliable and as close to the appropriate tissue as possible.

Discrepancies in the effect of flavonoids as SRT agents for MPS might arise from the experimental model chosen for in vitro or in vivo studies. Most of the results published to date were obtained for only one cell, tissue, or animal model type. Hence, in this study, we assessed the usefulness of two distinct cell culture models—murine fibroblasts and primary hepatocytes—to evaluate the in vitro effectiveness of flavonoids as SRT agents using the effects of genistein, kaempferol, and biochanin A on GAG synthesis in both cell types as an example.

It has been shown that genistein modulates various steps of the cell cycle, apoptosis, angiogenesis and metastasis, and possesses anti-inflammatory potential [38]. Many in vitro and in vivo studies have been performed since 2006, when Piotrowska et al. first described genistein as a potent inhibitor of GAG synthesis in MPS fibroblasts [13], followed by clinical usage of this isoflavone [12,15,27]. It has been suggested that the mechanism of genistein-mediated substrate reduction therapy involves the inhibition of EGFR phosphorylation [39]. Genistein was also shown to effectively modulate the expression of genes involved in GAG metabolism and the biogenesis of lysosomes [27], or to affect autophagy [40]. Most of these results were obtained from experiments using fibroblasts, but our comparison of the effects of genistein on GAG synthesis in two independent cell culture models revealed that the cellular response to genistein is very different for fibroblasts and hepatocytes. Genistein reduces the level of GAG synthesis in both fibroblasts and hepatocytes, but to our surprise, the effect is more significant in fibroblasts. It has been shown that genistein has a positive effect on lipid metabolism [41], and prevents such events as loss of cell viability and mitochondrial function, and the accumulation and peroxidation of triglycerides in a hepatocyte model of non-alcoholic fatty liver disease [42]. The dietary intake of genistein also suppresses hepatocellular carcinoma (HCC) through AMP-activated protein kinase (AMPK) activation and the suppression of inflammatory response in resident liver macrophages [43]. As multiple substrates were detected as primary or secondary storage materials in organs during lysosomal storage disorders, including MPS [44], the genistein-mediated modulation of GAG synthesis in hepatocytes may be a suitable model with which to study the phenomenon of secondary storage in MPS or the mechanisms underlying the resulting pathology.

Kaempferol and biochanin A are two other flavonoids whose activity as inhibitors of GAG synthesis is intensively studied [10,31]. The current study shows that kaempferol is even more effective than genistein in reducing the level of GAG synthesis in both fibroblasts and hepatocytes. In contrast, biochanin A shows only a slight inhibition in fibroblasts and does not affect hepatocytes. Kaempferol has many properties similar to genistein, including anti-inflammatory, antioxidant, and anticancer activity. The molecular mechanism of its action is based on the modulation of signalling pathways such as NF-κB, PI3K/AKT, MAPK, Bcl-2, Caspase-3, and VEGF [45]. It can also modulate the expression of GAG metabolism-related genes and significantly stimulate the expression of transcription factor EB (TFEB) in healthy and MPS II human fibroblasts [27]. No data is available to date on the effect of kaempferol on GAG metabolism in hepatocytes, but a significant effect on the accumulation of triglycerides in the liver was observed by inhibiting the Akt/mTOR pathway and inducing hepatic autophagy [46]. Our results indicate that, similarly to genistein, kaempferol can be effectively used as an inhibitor of GAG synthesis in SRT for MPS, but the detailed molecular mechanism of its action on GAG metabolism in MPS should be further investigated. Biochanin A has also been found to exhibit various pharmacological effects that have potential applications, including neuroprotective, anti-cancer, anti-oxidant, anti-inflammatory, and anti-hyperlipidaemia activity, which makes it a useful phytochemical for the treatment of cancer, neurodegenerative diseases, diabetes, menopausal symptoms, and osteoarthritis [47,48,49,50]. It also induces apoptosis in cancer cells and regulates the expression of enzymes involved in drug metabolism [51]. The beneficial effects of biochanin A result from affecting MAPK, NF-κB, TNF-α, TGF-b1 and PPAR γ [47,51]. Both genistein and biochanin A have a close structural similarity to the endogenous oestrogen 17β-oestradiol (E2) and act through nuclear oestrogen receptors (ER) [47]. Here we show that the effect of biochanin A on the inhibition of GAG synthesis in fibroblasts is comparable to that of genistein and kaempferol, but is in contrast to that obtained for hepatocytes, in which the inhibition was weaker compared to other flavonoids. Biochanin A is also a less effective GAG inhibitor when used in a two-component mixture with genistein. The beneficial effect of biochanin A, therefore, seems to be limited to only certain cell types. It confirms our statement that in vitro tests should always include many different types of cells. The low water solubility and poor bioavailability of biochanin A [52] also severely limit its clinical use. However, some studies show that the bioavailability of biochanin A may be increased by the simultaneous administration of other compounds. For example, the co-administration of quercetin increased the bioavailability of biochanin A by 3.44 times, and its urinary excretion by 2.97-fold [53].

Two active agents used in combination may have a stronger or weaker effect on cell metabolism. The main reason for using combination therapy is to increase the efficacy of the treatment while decreasing its toxicity. It has recently been shown that including genistein and kaempferol in the mixture results in synergy, additivity, or slight antagonism, and the type of interaction depends on the concentration and component ratios [31]. Increasing the amount of kaempferol in place of genistein also results in the less efficient inhibition of cell proliferation [31]. Our study reveals the entire spectrum of interactions that depend not only on the composition, concentration, and ratio of a mixture but also on the cell culture model used in the experiment. When genistein was used in combination with either kaempferol or biochanin A, we observed positive interactions such as synergy or additivity for fibroblasts, but antagonism for hepatocytes (see Figure 10). Our results clearly show that in GAG synthesis, the types of interactions between mixture components depend strongly on the concentration and type of cell. Such contradictory interactions of the same mixture in different cell types may be explained by the activity of cytochrome P450 (CYP) enzymes. CYPs located in the mitochondria and endoplasmic reticulum of hepatocytes are the major liver enzymes involved in the phase I metabolism of a wide range of endogenous compounds, as well as xenobiotics, including dietary products and drugs. Most clinical drugs and xenobiotics are metabolised by CYP3A4 [54], and dietary polyphenols can interact with this enzyme, altering its expression and activity [55]. For example, kaempferol inhibits, and genistein activates, this CYP isoform [56]. CYP enzymes are highly active in hepatocytes, as these cells play a major role in detoxification. However, only low levels of CYP mRNAs expression or low enzyme activity have been detected in human dermal fibroblasts (HDF) [57,58]. The ability of CYP enzymes to mediate the metabolism of oestradiol or isoflavones, including genistein, is well described [59,60]. Genistein is thought to be metabolised in the gastrointestinal tract and liver [61] which are both sites with abundant levels of CYPs. In general, the CYP-mediated metabolism of xenobiotics results in metabolites that can be less biologically active or more readily excreted than the substrates, and this group of enzymes may thus play a considerable role in the bioactivity of flavonoids by producing active or inactive metabolites. We assume that the ability of hepatocytes used in our study to metabolise xenobiotics persists during the first few days of in vitro culture, and since the rate of xenobiotic metabolism determines not only the toxicity but also the activity of a chemical, it can be expected that the biological activity of flavonoids will vary for different cell types. Here, we hypothesise that fibroblasts are less able to metabolise flavonoids than hepatocytes, which results in such remarked differences in response to genistein, kaempferol or biochanin A, or in the drug-drug interactions between these compounds that we observed with both cell culture models. Flavonoid metabolites produced by CYP enzymes may also affect the metabolism of other CYP substrates, which in turn can significantly change the type of drug-drug interactions observed, depending on the flavonoid mixture composition, concentration or ratio tested, and on the cell culture model used.

In summary, both fibroblasts and hepatocytes are good cell culture models for studying the molecular aspects of the pathophysiology of MPS I, and for conducting in vitro tests to evaluate the effectiveness of new therapeutics (Table 2 summarises the advantages and disadvantages of both models). However, it should be taken into account that to obtain a more complete answer to the research question, the analysis should be performed using more than one cell type.

## 4. Materials and Methods

### 4.1. Animal Husbandry

Animal maintenance and all the experimental procedures were conducted according to the guidelines for animal welfare and were ethically approved by the Polish Local Ethical Committee Protocol code no. 32/2011. Mice were housed under specific pathogen-free (SPF) conditions, in a 12/12h light/dark cycle, constant temperature, and humidity, with food and drink ad libitum. The MPS I knockout mouse model strain B6.129-Idua^tm1Clk^/J was purchased from the Jackson Laboratory (Bar Harbor, ME, USA). The breeding colony was established and maintained from heterozygous mating pairs. To establish the genotype, DNA from ear punches was isolated using QIAamp DNA Mini Kit (Qiagen, Manchester, UK) and used as a template to perform PCR with allele-specific oligonucleotides: IMR 1451 5′-GGAACTTTGAGACTTGGAATGAACCAG-3′, IMR 1452 5′-CATTGTAAATAGGGGTATCCTTGAACTC-3′, and IMR 1453 5′-GGATTGGGAAGACAATAGCAGGCATGCT-3′. Wild-type (WT) and mutant (MPS I) mice were used to derive fibroblasts and hepatocytes.

### 4.2. Murine Fibroblasts Derivation

Mice were sacrificed at the age of 6–8 weeks by cervical dislocation, and the tails were immediately cut off and placed in Dulbecco’s phosphate-buffered saline (DPBS; Thermo Fisher Scientific, Waltham, MA, USA) with a 2% antibiotic-antimycotic solution (A/A; Thermo Fisher Scientific, Waltham, MA, USA). Under sterile conditions, skin from the tail was removed and cut into 2–3 mm squares. Five to six skin pieces were placed in the centre area of a 6-well plate and allowed to air dry for 5 min. Tissue fragments were covered with a drop of culture medium (i.e., Dulbecco’s modified Eagle medium (DMEM; Thermo Fisher Scientific, Waltham, MA, USA) supplemented with 10% foetal bovine serum (FBS; Thermo Fisher Scientific, Waltham, MA, USA) and 1% A/A), followed by adding medium to the minimum level and placing in a humidified 37 °C, 5% CO_2_ incubator. Skin explants were cultured for 10 days, and when fibroblasts effectively migrated from the tissue, skin fragments were removed, and a fresh culture medium was added to each well. When cultures reached around 80% confluency, the medium was removed, and cells were washed with DMEM supplemented with 1% A/A without FBS and passaged for the first time.

### 4.3. Murine Hepatocytes Derivation

We developed a murine primary hepatocyte derivation protocol similar to that recently published [63]. Briefly, the day before hepatocyte isolation, plates for seeding cells were coated with collagen type I (Thermo Fisher Scientific, Waltham, MA, USA) at a concentration of 12.5 µg/cm^2^ well. Plates were incubated at room temperature for 1 h, washed three times with DPBS solution, allowed to air dry, and UV sterilised for 30 min in a laminar flow cabinet, then stored at 4 °C until use. At the age of eight weeks, mice were anaesthetised with ketamine (Biowet Puławy, Puławy, Poland) and xylazine (Biowet Puławy, Puławy, Poland), at doses of 150 and 20 mg/kg body weight, respectively. Induction of anaesthesia was acquired after the loss of the backing limb reflex. Mice were placed on the heating mat and immobilised. The abdomen was disinfected with alcohol before laparotomy. To start liver perfusion, the tip of a wing needle was introduced into the inferior vena cava and the Liver Perfusion Medium (LPM; Thermo Fisher Scientific, Waltham, MA, USA) was flowed using a peristaltic pump with a constant speed of 7 mL/min and temperature of 37 °C. The hepatic portal vein was cut to obtain a free flow of LPM. When the liver was drained of blood until it appeared pale, the LPM was changed to Liver Digest Medium (LDM; Thermo Fisher Scientific, Waltham, MA, USA). The location of the cut in the hepatic portal vein was temporarily clamped to increase the effectiveness of digestion. Perfusion was retained until the organ lost its compact structure, then, the liver was cut out and transferred to a sterile tube with a 4 °C L-15 Medium (Thermo Fisher Scientific, Waltham, MA, USA). Under sterile conditions, on ice, the liver was transferred to a Petri dish containing 20 mL LDM and chopped. The cell suspension was transferred to a Falcon tube and centrifuged at 50× *g* at 4 °C for 5 min. The supernatant was discarded, and the pellet was resuspended in 15 mL of Hepatocyte Wash Medium (HWM; Thermo Fisher Scientific, Waltham, MA, USA) and filtered three times with a 100-µm cell strainer followed by centrifugation at 50× *g* at 4 °C for 5 min. The pellet was resuspended in 10 mL of HWM and centrifuged on Percoll with a density of 1.08 g/mL (Sigma-Aldrich, Darmstadt, Germany at 750× *g*, at 20 °C for 20 min. The cells pellet was resuspended in 10 mL HWM and centrifuged twice at 50× *g* for 10 min. Finally, cells were resuspended in 5 mL of William’s E Medium (Thermo Fisher Scientific, Waltham, MA, USA) supplemented with 5% FBS and 10% A/A solution. Viable hepatocytes were plated in the collagen pre-coated plates at densities of 5 × 10^5^ cells/well in 6-well plates or 5 × 10^4^ cells/well in 48-well plates. Cells were left to adhere for 3 h in a humidified 37 °C, 5% CO_2_ incubator, and then the medium was changed to HepatoZYME-SFM (Thermo Fisher Scientific, Waltham, MA, USA) supplemented with 1.25 µg/cm^2^ collagen. After 24 h, the medium was changed to HepatoZYME-SFM supplemented with 1% 200 mM L-glutamine (Thermo Fisher Scientific, Waltham, MA, USA) and 1% A/A. The medium was changed to fresh every 48 h during culture.

### 4.4. Cell Viability Assay

The viability of cells was estimated using the Muse™ Count & Viability Reagent (Merck Millipore, Billerica, MA, USA) on the Muse™ Cell Analyzer (Merck Millipore, Billerica, MA, USA) according to the standard protocol.

### 4.5. Measurement of GAG Content

Recently, it was shown that the effect of genistein on the amount of GAG depends on the cell types [64]. The level of GAG was therefore determined in fibroblasts, which are the most frequently chosen in vitro model in this type of study, and in hepatocytes—cells whose metabolism is significantly disturbed in the course of MPS I and which are involved in metabolizing drugs. Fibroblasts were plated to a final density of 1 × 10^5^ per well in 6-well plates and cultured in full standard medium (DMEM supplemented with 10% FBS and 1% A/A) which was changed every 3–5 days depending on the culture confluence. Hepatocytes were plated to a final density of 5 × 10^5^ per well in 6-well plates, as described above. At the endpoint of the experiment, cells were collected and digested with 0.5% papain (Sigma Aldrich, Darmstadt, Germany) in 200 mM PBS, pH 6.4 for 3 h at 65 °C. Total GAG content was estimated with a Blyscan kit (Biocolor Ltd., Carrickfergus, UK) and normalised per DNA concentration assessed with the Quant-iT™ PicoGreen^®^dsDNA Reagent (Thermo Fisher Scientific, Waltham, MA, USA) according to the manufacturer’s protocols.

### 4.6. Flavonoid Solutions

Kaempferol (3, 5, 7-trihydroxy-2-(4-hydroxyphenyl) chromen-4-one) and biochanin A (5, 7-dihydroxy-3-(4-methoxyphenyl) chromen-4-one) were purchased from AK Scientific, Inc. (Union City, CA, USA). Genistein (5, 7-dihydroxy-3-(4-hydroxyphenyl) chromen-4-one) was synthesised by the Pharmaceutical Research Institute (Warsaw, Poland). Stock solutions of flavonoids were dissolved in dimethyl sulfoxide (DMSO; Sigma-Aldrich, Darmstadt, Germany) and were stored at −20 °C, in the dark. Working solutions of flavonoids were freshly prepared at appropriate concentrations in DMEM supplemented with 10% FBS and 1% A/A solution for fibroblasts, or in HepatoZYME with 1% A/A, without FBS for hepatocytes. Mixtures of genistein and kaempferol, and genistein and biochanin A, were prepared in total concentrations of 30 and 60 µM and at ratios of 4:1, 3:2, 2:3, and 1:4 for the indicated compounds.

### 4.7. Measurement of GAG Synthesis Kinetics

The kinetics of GAG synthesis in cell cultures was assessed by measuring the incorporation of glucosamine, D-[1-3H] hydrochloride (Hartmann Analytic GmbH, Braunschweig, Germany)—refereed in the text as 3H-GlcN—into GAG chains. Fibroblasts were plated to a final density of 2 × 10^4^ per well in 48-well plates in DMEM supplemented with 10% FBS and 1% A/A solution and incubated overnight. Hepatocytes were plated to a final density of 5 × 10^4^ per well in 48-well plates in HepatoZyme supplemented with 5% FBS. The medium was changed after 4 h to HepatoZYME supplemented with 1.25 µg/cm^2^ collagen and next to HepatoZyme without FBS, as described above. Following overnight incubation, cells were exposed to appropriate concentrations of single flavonoids or flavonoid mixtures for 24 h. Control cells were exposed to 0.05% DMSO under the same conditions. Cells were then labelled with 10 µCi/mL of 3H-GlcN in DMEM without glucose and pyruvate (Sigma-Aldrich, Darmstadt, Germany) supplemented with 1% A/A and 10% or 3% FBS for fibroblasts or hepatocytes, respectively, and single flavonoids, their mixtures, or DMSO alone (control), for an additional 24 h. After labelling, cells were washed six times with DPBS. Next, fibroblasts were digested with 0.5% papain (prepared in 200 mM phosphate buffer, pH 6.4) for 3 h at 65 °C. Hepatocytes were first digested with collagenase (prepared in Earle’s balanced salt solution) for 40 min at 37 °C, then with 1% papain (prepared in 400 mM DPBS) for 3 h at 65 °C. The incorporation of 3H-GlcN in papain-digested samples was measured in MicroBeta2 scintillation counter (PerkinElmer, Waltham, MA, USA), and the DNA concentration was quantified with Quant-iT™ PicoGreen^®^dsDNA Reagent according to the manufacturer’s protocol. The incorporation of 3H-GlcN (counts per minute; cpm) was calculated relative to DNA amount and expressed as cpm/ng of DNA, and the efficiency of GAG synthesis in flavonoid-treated cultures was normalised against that of control cells.

### 4.8. Data Analysis

Statistical differences in GAG content and GAG synthesis level were established with a two-tailed *t*-test, and the effect of flavonoid mixtures on GAG synthesis was estimated using one-way ANOVA with Tukey’s post-hoc test. Significance was defined as *p* < 0.05. All tests were performed using Statistica 13.1 (StatSoft, Kraków, Poland). Analyses of interactions between flavonoids in two-component mixtures, including calculation of combination index (CI) and dose reduction index (DRI), construction of respective plots, and normalised isobolograms were performed using CompuSyn 1.0 software (ComboSyn, Inc., Paramus, NJ, USA). The assumption of the analysis was the non-constant ratio for combinations with the inclusion of data for single flavonoids.

## Figures and Tables

**Figure 1 ijms-24-00534-f001:**
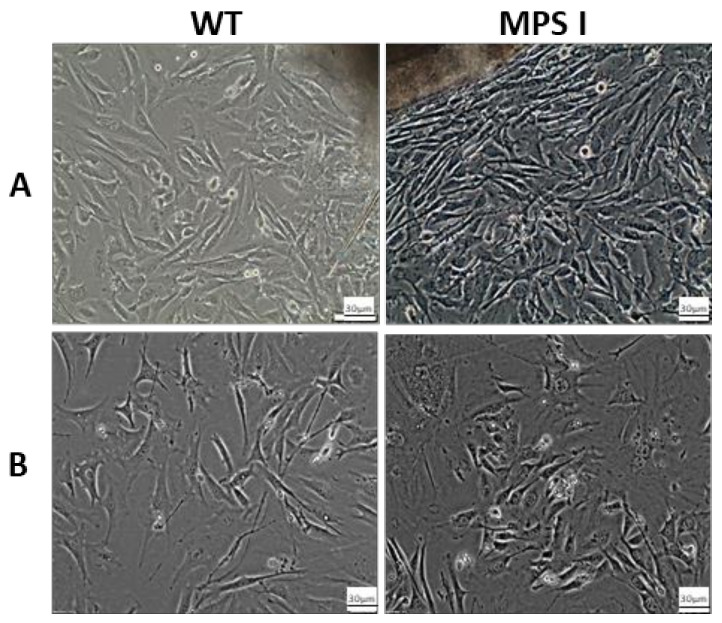
Morphology of wild type (WT) and mucopolysaccharidosis I (MPS I) murine fibroblasts. Microphotographs represent different time points during cell culture. Panel (**A**) show fibroblasts that migrated from the skin explants on Day 10 after the beginning of culture. Panel (**B**) show murine fibroblasts after the fourth passage.

**Figure 2 ijms-24-00534-f002:**
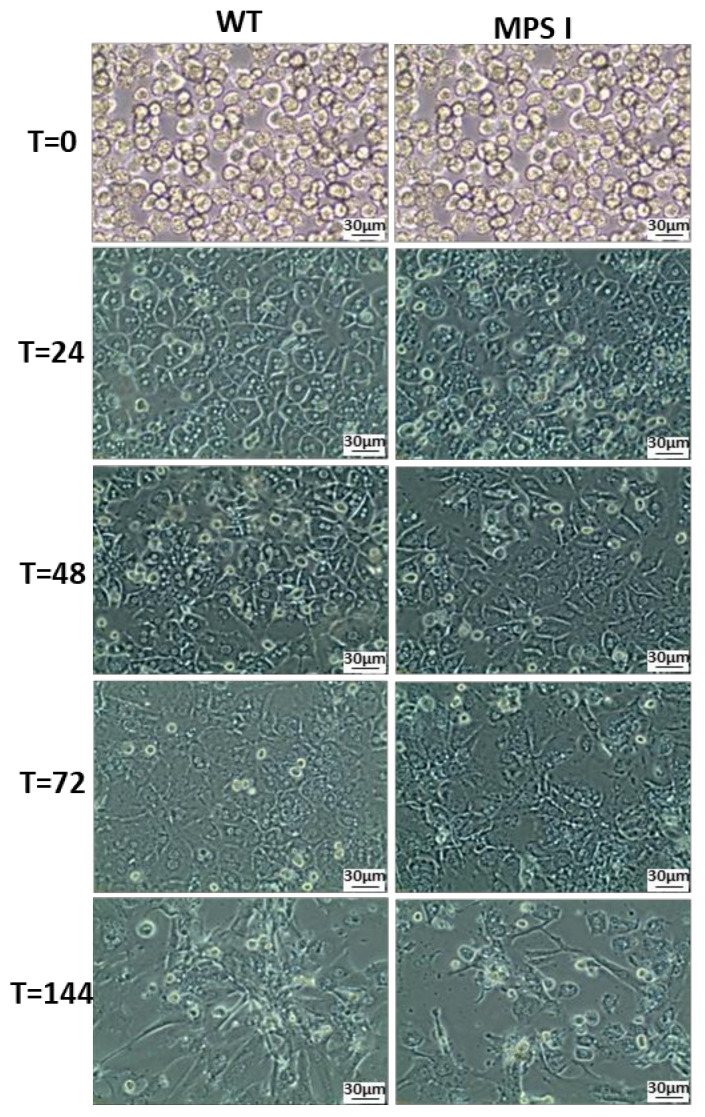
Morphology of wild type (WT) and mucopolysaccharidosis I (MPS I) murine primary hepatocytes. Microphotographs show cell morphology at several time points during cell culture. T = 0 hepatocytes at the beginning of culture when they do not attach to the bottom of the dish and after 24, 48, 72, and 144 h of culture.

**Figure 3 ijms-24-00534-f003:**
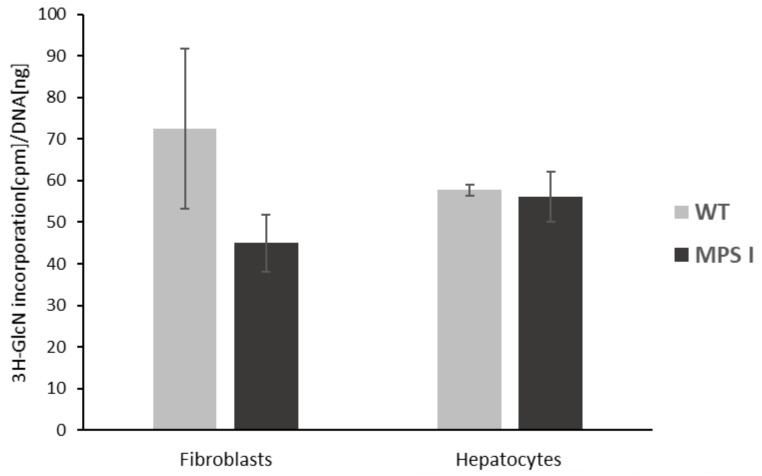
Kinetics of glycosaminoglycans (GAG) synthesis in murine fibroblasts and primary hepatocytes. The incorporation of radioactively labelled glucosamine (3H-GlcN) into GAGs was measured after 24 h of culture in a scintillation counter and calculated according to the amount of DNA. Light grey bars represent the level of GAG synthesis in wild-type (WT) cells, and dark grey bars—in mucopolysaccharidosis I (MPS I) cells. Results are presented as average values ± SD for two separate experiments and biological repeats: N = 5 for fibroblasts and N = 6 for hepatocytes. No statistical significance was detected by the *t*-test.

**Figure 4 ijms-24-00534-f004:**
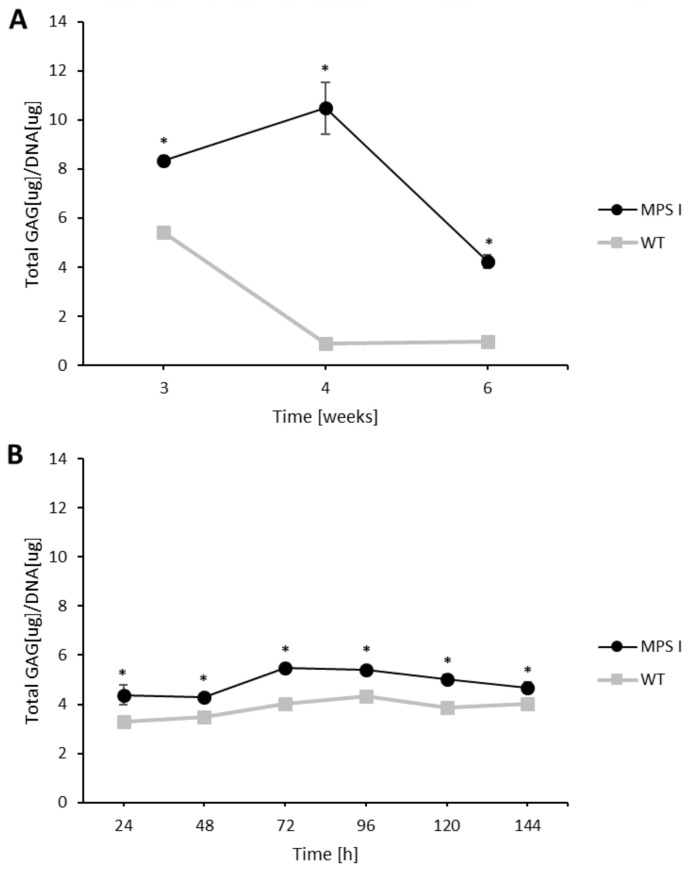
Total GAG content in mucopolysaccharidosis I (MPS I, black circles) and wild type (WT, grey squares) murine fibroblasts (**A**) and primary hepatocytes (**B**) during cell culturing. Results are presented as the mean ± SD obtained for three different cell cultures for two different experiments in each case. An asterisk (*) indicates statistically significant differences between MPS I and WT cells (two-tailed *t*-test, *p* < 0.05).

**Figure 5 ijms-24-00534-f005:**
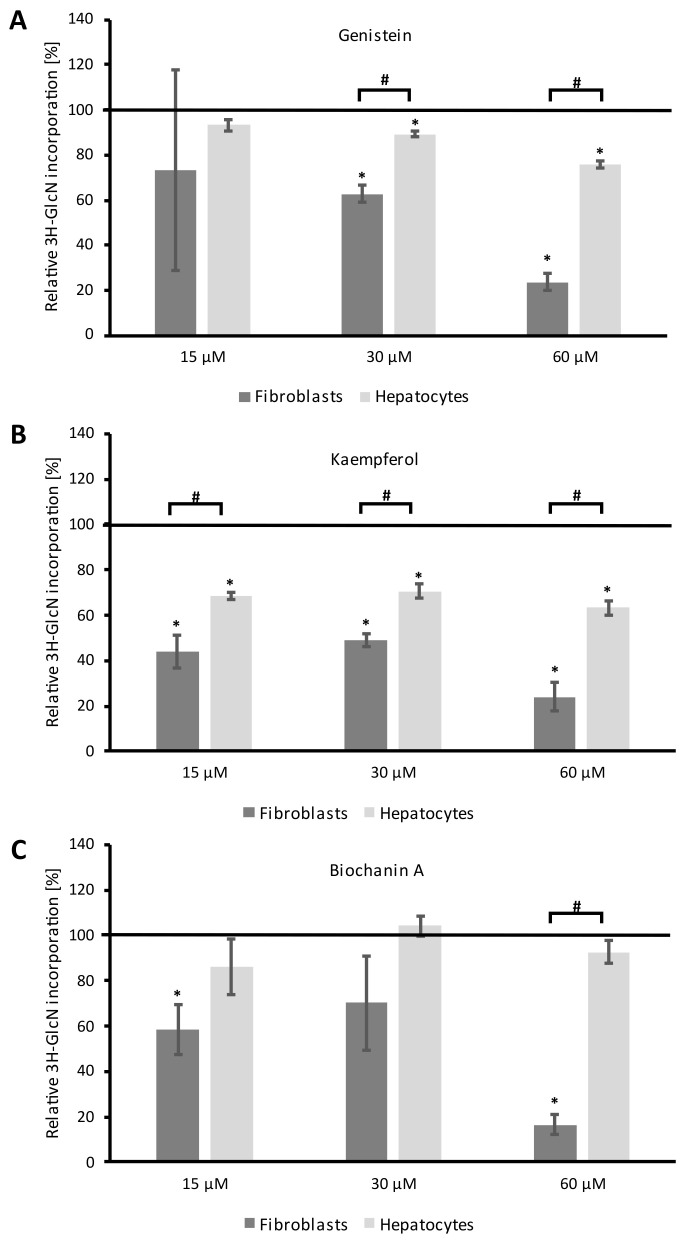
The level of GAG synthesis in fibroblasts and primary hepatocytes after 48 h of exposure to single flavonoids: genistein (**A**), kaempferol (**B**), and biochanin A (**C**) at final concentrations of 15 µM, 30 µM, and 60 µM respectively, measured by D-[1-3H]-glucosamine incorporation assay. The black line (100%) corresponds to metabolic activity in control cells treated with 0.05% DMSO. Bars represent the average ± SD obtained for two different cell cultures from two different experiments. An asterisk (*) indicates statistically significant differences from control cells (ANOVA with Tukey’s post hoc, *p* < 0.05); hashtag (#) indicates statistically significant differences between cell types (*t*-test, *p* < 0.05).

**Figure 6 ijms-24-00534-f006:**
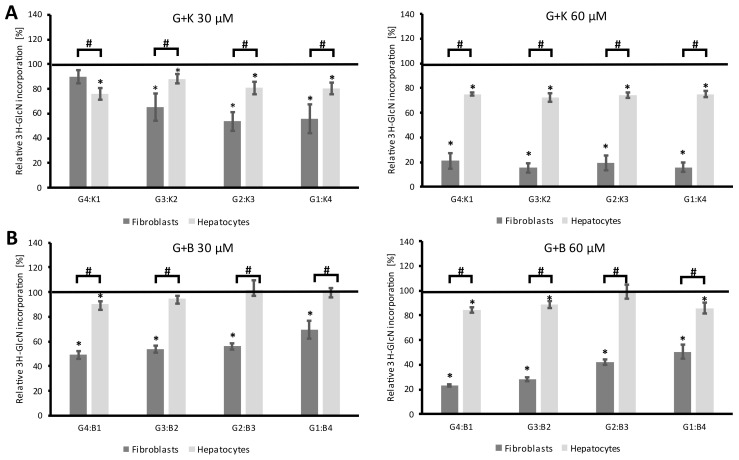
The level of GAG synthesis in fibroblasts and primary hepatocytes after 48 h of exposure to flavonoids in two-component mixtures: genistein and kaempferol (**A**), and genistein and biochanin A (**B**) at final concentrations of 30 µM and 60 µM, respectively, measured by the D-[1-3H]-glucosamine incorporation assay. The black line (100%) corresponds to metabolic activity in control cells treated with 0.05% DMSO. Bars represent the average ± SD obtained for two different cell cultures from two different experiments. An asterisk (*) indicates statistically significant differences from control cells (ANOVA with Tukey’s post hoc, *p* < 0.05); hashtag (#) indicates statistically significant differences between cell types (*t*-test, *p* < 0.05). G—genistein, B—biochanin A, K—kaempferol.

**Figure 7 ijms-24-00534-f007:**
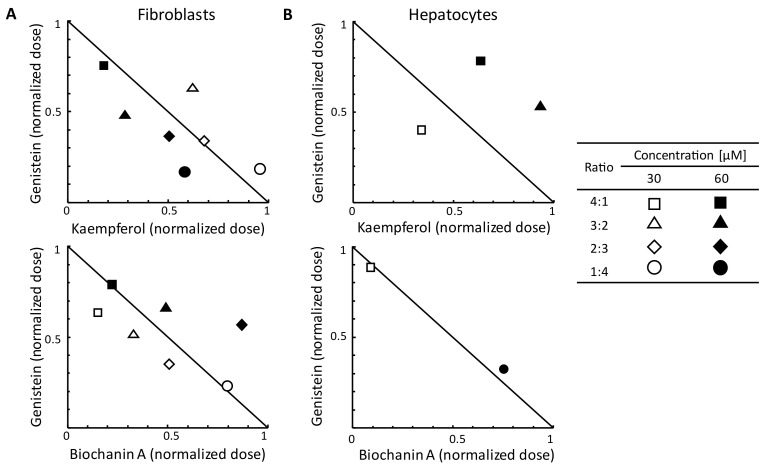
Normalised isobolograms for the non-constant ratio combinations of genistein and biochanin A, and genistein and kaempferol regarding the effect on GAG synthesis inhibition in fibroblasts (**A**) and hepatocytes (**B**). The normalised doses for each component drug were calculated as [D/Dx], where D means the dose of a drug and Dx means the dose of that drug used alone to inhibit the GAG synthesis by x%; this results in values scaled from 0 to 1 for both, x- and y- axes. The solid hypotenuse line denotes the theoretical additive effect.

**Figure 8 ijms-24-00534-f008:**
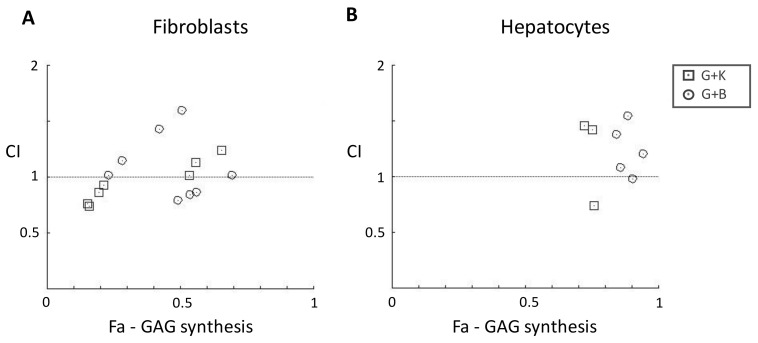
Combination index (CI) plot. CI values for various drug ratios are plotted against the fraction affected (Fa) values, i.e., the fraction of GAG synthesis inhibition obtained for fibroblasts (**A**) and hepatocytes (**B**). The dotted horizontal line represents the additive effect. CI values from actual data points are indicated by symbols. Fa—fraction affected, CI—combination index, G—genistein, B—biochanin A, K—kaempferol.

**Figure 9 ijms-24-00534-f009:**
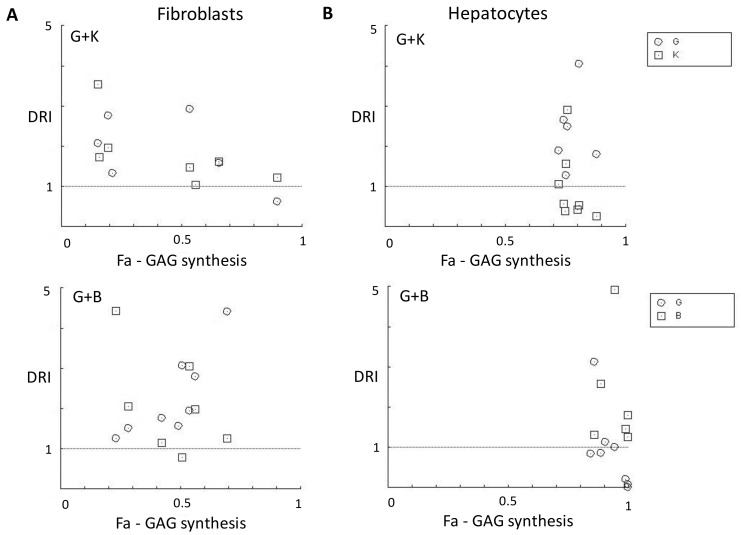
The dose reduction index (DRI) plots. DRI values are plotted against the fraction affected (Fa) values, i.e., the fraction of GAG synthesis inhibition after treatment of fibroblasts (**A**) and hepatocytes (**B**) with mixtures of genistein and biochanin A (**top graphs**) or genistein and kaempferol (**bottom graphs**). The dotted horizontal line represents no dose reduction; points plotted with DRI > 1 show favourable dose reduction and with DRI < 1—unfavourable dose reduction. Fa—fraction affected, DRI—dose reduction index, G—genistein, B—biochanin A, K—kaempferol.

**Figure 10 ijms-24-00534-f010:**
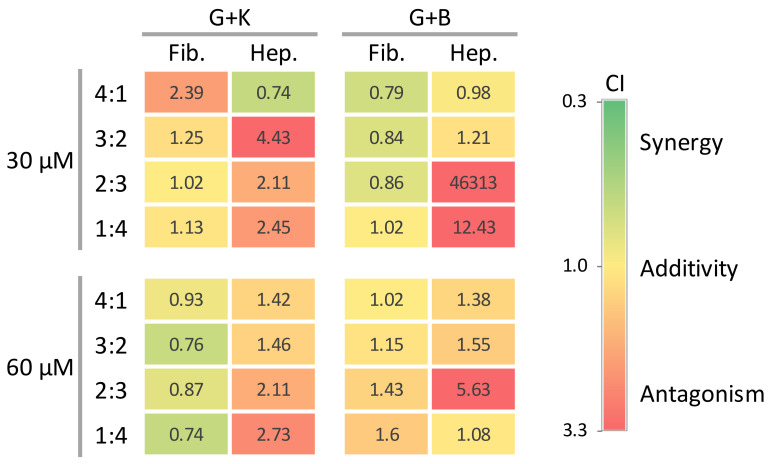
Interactions between flavonoids in two-component mixtures regarding the effect on GAG synthesis in fibroblasts and hepatocytes. The heat map shows colour-coded combination index values calculated from experimental data obtained for genistein mixed with either biochanin A or kaempferol at non-constant ratios (4:1, 3:2, 2:3, and 1:4) and total flavonoid concentrations of 30 and 60 µM. Fib.—fibroblasts, Hep.—hepatocytes, B—biochanin A, G—genistein, K—kaempferol, CI—combination index.

**Table 1 ijms-24-00534-t001:** Dose reduction index (DRI) for mixtures of genistein and kaempferol, and genistein and biochanin A regarding the effect on the inhibition of GAG synthesis in fibroblasts and hepatocytes. G—genistein, B—biochanin A, K—kaempferol.

		G + K				G + B			
		Fibroblasts		Hepatocytes		Fibroblasts		Hepatocytes	
Total Conc.	Ratio	G	K	G	K	G	B	G	B
30 µM	4:1	0.63	1.23	2.49	2.91	1.57	6.41	1.13	10.18
	3:2	1.59	1.62	1.81	0.26	1.95	3.04	1.00	4.91
	2:3	2.92	1.48	4.05	0.54	2.80	1.98	0.00	1.26
	1:4	5.64	1.05	8.28	0.43	4.42	1.26	0.08	1.81
60 µM	4:1	1.33	5.54	1.28	1.57	1.27	4.43	0.84	5.26
	3:2	2.08	3.54	1.90	1.07	1.51	2.05	0.86	2.57
	2:3	2.77	1.97	2.65	0.58	1.77	1.15	0.20	1.46
	1:4	6.13	1.73	5.13	0.39	3.07	0.79	3.13	1.31

**Table 2 ijms-24-00534-t002:** Summary of advantages and disadvantages of fibroblasts and hepatocytes as a model for studying the pathophysiology of mucopolysaccharidosis I (MPS I) and testing potential therapeutics.

Cell Type	Advantages	Disadvantages
Fibroblasts	Easy to obtain from humans and animals Simple protocol for derivation Cells proliferate in vitro Experimental window up to passage 4 (good viability) Results are mainly obtained for unmetabolised xenobiotics	Variable GAG ^1^ storage levels depend on the duration of cultivation. Low expression and activity of CYP ^2^ enzymes [57,58] Aspects of xenobiotic metabolism are covered to some extent or not at all
Hepatocytes	Stable GAG storage levels during cultivation Good viability up to 6 days Stable MPS I phenotype between Day 3 and Day 6 High expression and activity of CYP enzymes [62] Aspects of xenobiotic metabolism are well addressed	More elaborate and multistep derivation protocol Difficult to obtain from humans Short experimental window (up to 6 days) Cells do not proliferate in vitro

^1^ GAG—glycosaminoglycans; ^2^ CYP—cytochrome P450 enzymes.

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
