# Peer review of "Murine Fibroblasts and Primary Hepatocytes as Tools When Studying the Efficacy of Potential Therapies for Mucopolysaccharidosis Type I"

_ijms, 2022, doi:10.3390/ijms24010534_

Round 1

Reviewer 1 Report

This is a good study to find potential therapeutic agents of a genetic disorder. The result and discussion sections were nicely elaborated. Introduction section is also fine but if one or two recent studies in which flavonoids were used as  therapeutic agents for this disease will be added it will make it perfect.  

Author Response

Esteemed Reviewers,

We are grateful for the efforts made during the evaluation of our manuscript, as well as for suggestions and recommendations, which lead to the improvement of the quality of our article. Please find below our point-by-point answers (in red) to your remarks.

#Reviewer 1 Comments and Suggestions for Authors

>Introduction section is also fine but if one or two recent studies in which flavonoids were used as  therapeutic agents for this disease will be added it will make it perfect.”

Answer: We emphasize more recent research on flavonoids in MPS therapy and have added information on the latest research on flavonoid encapsulation to increase their effectiveness and on the search for new sources of these compounds (lines 90-93, refs. 25 and 26).

Reviewer 2 Report

The manuscript of Wesierka et al's describes a protocol to obtain an in vitro model of MPS1 mouse hepatocytes and fibroblasts and the effect of flavonoids on the synthesis and accumulation of GAGs. The manuscript is original and well structured.

Minor revisions:

Introduction:

I recommend entering some data on the clinical symptoms and signs of mucopolysaccharidosis 1

materials and methods:

Indicate why fibroblasts and hepatocytes were chosen as in vitro cell line

Discussion

The section relating to the possibility of SRTs to cross the BEE should be anticipated in the introduction

Author Response

Esteemed Reviewers,

We are grateful for the efforts made during the evaluation of our manuscript, as well as for suggestions and recommendations, which lead to the improvement of the quality of our article. Please find below our point-by-point answers (in red) to your remarks.

#Reviewer 2 Comments and Suggestions for Authors

Minor revisions:

>Introduction: I recommend entering some data on the clinical symptoms and signs of mucopolysaccharidosis 1

Answer: We have inserted the information on the clinical symptoms of mucopolysaccharidosis I in the Introduction: lines 42-47 with reference nr 2. We also included information on different therapeutic approaches and their impact on clinical symptoms (lines 48-53) and added ref. nr 3 and 4.

>Materials and methods: Indicate why fibroblasts and hepatocytes were chosen as in vitro cell line

Answer: In the Materials and Methods section, we have explained why hepatocytes and fibroblasts were selected as in vitro cell models: lines 521-525.

>Discussion

The section relating to the possibility of SRTs to cross the BBB should be anticipated in the introduction.

Answer: We have transferred the fragment about crossing the BBB from the Discussion to the Introduction: lines 63-73.
